# Identification of Reassortment of *Orthotospovirus citrullomaculosi* in Jiangxi Province, China

**DOI:** 10.3390/v17111448

**Published:** 2025-10-31

**Authors:** Bin Peng, Xinlong Zhang, Na Cao, Chengpu Yan, Fangshu Li, Fanghong Zhu

**Affiliations:** Jiangxi Key Laboratory of Horticultural Crops (Fruit, Vegteable & Tea) Breeding, Institute of Horticultural Sciences, Jiangxi Academy of Agricultural Sciences, Nanchang 330200, China; pengbin@jxaas.cn (B.P.);

**Keywords:** *Citrullus lanatus*, high-throughput sequencing, sequence alignment, phylogenetics, reassortment, gene flow, *Orthotospovirus citrullomaculosi*

## Abstract

Watermelon silver mottle virus (WSMoV) is a thrips-transmitted *Orthotospovirus* that severely impacts cucurbits production across Asia. Although previous diversity studies focused on the nucleocapsid (*N*) gene, genome-level evolutionary analyses are lacking. In 2023 and 2024, symptomatic watermelon in Jiangxi Province, China, was analyzed by RT-PCR and high-throughput sequencing, yielding complete genomes of two WSMoV isolates, FZNC and THBC. Multiple-sequence alignment and phylogenetic analysis of the complete sequences of L, M, and S RNAs defined two phylogenetic clades (O and N). However, the Jiangxi isolates clustered in different clades for the L segment versus the M and S segments, suggesting a potential reassortment event. This conclusion was confirmed by RDP4 and RT-PCR analysis, which identified a significant reassortment event involving an L RNA segment derived from a Guangdong isolate (Clade N) and the M and S segments from a Taiwan isolate (Clade O). This study provides the first evidence of natural reassortment in WSMoV, underscoring its potential for rapid evolution. It also constitutes the first report of WSMoV in Jiangxi Province, in East China, marking a concerning expansion of its geographic range into inland China and raising the risk of cucurbit production.

## 1. Introduction

Watermelon silver mottle virus (WSMoV; species name *Orthotospovirus citrullomaculosi*) belongs to the family *Tospoviridae*, genus *Orthotospovirus*. In the field, it is primarily transmitted by the melon thrips (*Thrips palmi*). The viral genome consists of three segmented single-stranded RNAs, designated L, M, and S. The L RNA is negative-sense and contains a single large open reading frame (ORF) encoding the RNA-dependent RNA polymerase (RdRp). The M RNA is ambisense and encodes two ORFs: the ORF on the sense strand encodes the non-structural movement protein NS_M_, and the ORF on the complementary-sense strand encodes the precursor glycoprotein (G1/G2). The S RNA is also ambisense and encodes two ORFs: the ORF on the sense strand encodes the non-structural protein NS_S_, and the ORF on the complementary-sense strand encodes the viral nucleocapsid protein (N) [1]. Species demarcation within Tospoviridae is based on >10% amino acid divergence in the N protein. WSMoV was initially considered a watermelon strain of tomato spotted wilt virus but was later identified as a distinct species within the same genus (*Orthotospovirus*) based on sequencing of the *N* gene located on the S RNA segment [1,2].

To date, WSMoV has only been reported in Asia. It was first recorded on watermelon in Japan in 1982 [2]; subsequently, there were outbreaks on watermelon in Taiwan, China, in 1992 [3], and it was detected on watermelon, pepper, tomato, and *Physalis heterophylla* in Thailand in 2007 [4]. The virus was first reported in mainland China on watermelon in Guangdong Province in 2011 [5], followed by successive reports on various hosts: pepper in Guangdong [6]; watermelon in Yunnan [7]; watermelon and melon in Guangxi [8]; pepper in Guangxi [9]; cucumber, squash, and whitegourd in Guangdong [10]; watermelon and melon in Hainan [11]; watermelon and cucumber in Shandong; watermelon in Zhejiang [12]; *Siraitia grosvenorii* in Guangxi [13]; and peanut in Yunnan [14].

On watermelon, WSMoV primarily causes silver mottling or mosaic on leaves [2]; early infection can lead to severe symptoms such as stunting and leaf deformation [3]. On melon, infection produces necrotic spots, while in pepper, the infection presents as chlorotic ring spots followed by necrosis [6]. WSMoV infects a range of economically important crops, especially cucurbits, and infections, particularly when they occur early in plant development, commonly reduce yield and marketability, causing substantial economic losses [2]. During the first decade after its detection in mainland China [5,6,7,8,9,10,11], WSMoV was mainly observed in coastal and tropical regions. However, over the past three years, it has been reported in inland provinces, including Shandong [12,15] and Hebei Province (isolate SJZ), consistent with a trend toward expansion into inland and temperate regions. Current genomic sequence analyses and phylogenetic studies of WSMoV mostly focus on the *N* gene [8,11,12,15], because this gene is the standard for species demarcation in the genus *Orthotospovirus*. Analyses based on complete genome sequences remain scarce but are important for clarifying population structure and understanding range expansion. In 2023 and 2024, watermelon plants exhibiting silver-gray mottling and leaf deformation, symptoms characteristic of WSMoV infection, were observed in fields in Jiangxi Province. This study collected suspected symptomatic samples and identified the occurrence of WSMoV on watermelon in Jiangxi Province using RT-PCR and HTS methods. The complete genome sequences of two isolates from Jiangxi Province were obtained. Sequence alignment, phylogenetic, and recombination/reassortment analyses were performed on the Jiangxi isolates. The first identification of WSMoV in Jiangxi Province extends the known geographic distribution of this virus and underscores its potential risk of spread into inland China, where it poses a threat to cucurbit crop production. Furthermore, phylogenetic analysis based on the nucleotides of complete genome and detection of reassortment events provides new insights into the origin and evolution of WSMoV.

## 2. Materials and Methods

### 2.1. Virus Disease Investigation and Sampling

In May 2023, watermelon leaves showing typical viral symptoms (mottling and deformity) were observed in a 0.67-hectare plastic tunnel in Jianchang Town, Nancheng County, Jiangxi Province. Similarly, in September 2023, mottling symptoms were observed on watermelon leaves in a hectare plastic tunnel in Nanxi Town, Taihe County, Jiangxi Province. Incidence was estimated using a random five-point sampling method: at each point, 10 plants were surveyed, and the average incidence rate was calculated from the five observations. During a disease investigation in the spring of 2024, we observed sporadic occurrences of symptoms suspected to be caused by WSMoV in several locations, Toubei Town (Guangchang County), Xianxia Town (Yudu County), and Xijiang Town (Huichang County), all within Jiangxi Province. Representative symptomatic samples were collected and stored at −80 °C in an ultra-low-temperature freezer.

### 2.2. Total RNA Extraction and RT-PCR Detection

Total RNA was extracted from diseased samples using the RNAsimple Total RNA Kit (Tiangen Biotech, Beijing, China). The concentration of the extracted and purified RNA was measured using a Nanodrop ND-1000 micro-UV-Vis spectrophotometer (Thermo Fisher Scientific, Waltham, MA, USA). RT-PCR detection was performed using the HiScript II One Step RT-PCR Kit (Vazyme Biotech, Nanjing, China). A 25 µL reaction system contained approximately 100 ng of RNA, with other components strictly following the kit instructions. The primers WS-F and WS-R (Appendix A) were used to detect WSMoV, which amplified a region of the *N* gene at an annealing temperature of 58 °C. Since cucumber green mottle mosaic virus (CGMMV) infection on watermelon produces symptoms similar to WSMoV, CGMMV was also detected in this study using primers CGCPF and CGCPR (Appendix A) with an annealing temperature of 56 °C. RT-PCR products, mixed with Ultra GelRed nucleic acid dye, were electrophoresed on 1% agarose gel. Target bands of the expected size were selected and sent to Sangon Biotech (Shanghai, China) for Sanger sequencing to obtain the nucleotide sequences.

### 2.3. High-Throughput Sequencing for Complete Genome Determination

Two symptomatic samples, FZNC1 (from Nancheng County) and THBC1 (from Taihe County), were selected for ribosomal RNA-depleted library construction and sequencing. The library construction and sequencing method followed that of Peng [16], briefly described here. Total RNA passing quality control was treated with the Ribo-ZeroTM Magnetic Kit (Illumina, San Diego, CA, USA) to deplete ribosomal RNA. Subsequently, libraries were constructed using the NEBNext Ultra Directional RNA Library Prep Kit (NEB, Ipswich, MA, USA). Libraries passing quality control were submitted to the Illumina NovaSeq platform for paired-end mode, yielding approximately 10 GB of data per sample. Raw sequencing data were processed to remove adapters and low-quality sequences to obtain clean data. Clean data were aligned against the watermelon (97103 v2) reference genome [17] to remove host-derived sequences. The remaining data were submitted to VirusDetect V1.7 [18] for viral genome sequence assembly and annotation. To obtain high-quality complete viral genomes, the transcriptome data were also assembled de novo using the RNA viral assembly module (--rnaviral) of SPAdes V4.2 [19], and the target virus genome sequences were extracted by BLASTn V2.17.

### 2.4. Multiple-Sequence Alignment and Phylogenetic Analysis

Complete L, M, and S segments for isolates FZNC and THBC were assembled and combined with GenBank-retrieved sequences: 8 full-length L, 8 full-length M, and 15 full-length S sequences (details shown in Table 1), plus 37 *N* gene sequences. Multiple-sequence alignments and clustering analyses were performed using SDTv1.3 to compute pairwise sequence identities for each genomic segment [20]. Phylogenetic analysis was performed using IQtree V2 software [21]; the optimal nucleotide substitution model was automatic selected, and maximum likelihood trees were constructed (with bootstrap support based on 1000 replicates) and visualized in iTOL V7.2 (https://itol.embl.de/).

### 2.5. Recombination and Reassortment Analysis

Recombination screening was performed in RDP4 [22] on full-length L, M, and S nucleotide sequences. Events detected by four or more of the seven methods implemented (RDP, GENECONV, BootScan, MaxChi, Chimaera, SiScan, 3Seq) with an average *p*-value < 0.001 were considered putative recombinants. For isolates showing no intra-segment recombination, L, M, and S were concatenated (L–M–S), and the concatenated sequences were re-tested in RDP4; recombination signals occurring exactly at segment junctions were taken as indicative of reassortment.

### 2.6. Validation of Reassortment

To validate the putative reassortment event, we designed segment-specific primer pairs for the L (WS-L4826F/WS-L5869R) and M (WS-M2347F/WS-M3278R, details in Appendix A) segments and used them together with the previously described S-segment primers (WS-F/WS-R) to amplify genomic fragments from seven samples collected in 2023 and five samples collected in 2024. Amplicons were Sanger-sequenced, and the resulting sequences were compared with the corresponding regions of the FZNC isolate.

## 3. Results

### 3.1. Occurrence and Symptoms

Infected watermelon plants primarily exhibited silver-gray mottling on the margins of older leaves (Figure 1A,B; in severe cases, the mottling covered more than 80% of the leaf area (Figure 1C). Younger leaves occasionally displayed marginal silver-gray mottling accompanied by leaf deformation (Figure 1D). The estimated disease incidence in Nancheng County and Taihe County was 46% and 24%, respectively.

### 3.2. RT-PCR Detection

Four symptomatic samples from Nancheng County and three from Taihe County were screened by RT-PCR using primers specific for WSMoV and CGMMV. All seven samples yielded the expected ~1000 bp amplicon for WSMoV (Appendix A), while no CGMMV-specific products were detected. All seven WSMoV amplicons were sequenced, producing fragments of 894 nt each; these two sequences shared 99.5–100% nucleotide identity. BLASTn searches returned the highest identities (99.22% and 99.44%, respectively) to the GL-1 WSMoV isolate from *Siraitia grosvenorii* in Guangxi. Collectively, the RT-PCR and sequencing results indicate that WSMoV is the primary causal agent of the observed silver-mottle disease in the surveyed watermelon fields in Nancheng and Taihe Counties. We applied the same WSMoV-specific RT-PCR assay to five suspected WSMoV samples (two in Guangchang County, one in Yudong County, and two in Huichang County) collected in 2024. Sanger sequencing of the amplicons and alignment to the homologous regions of FZNC1 revealed nucleotide identities >99.3% for all samples.

### 3.3. Complete Genome of WSMoV

Virus detection from high-throughput sequencing reads of samples FZNC1 and THBC1 was performed with VirusDetect. In addition to WSMoV, *Cucumis melo* amalgavirus 1 (CMAV1) was identified in FZNC1; THBC1 additionally contained CMAV1, melon aphid-borne yellows virus (MABYV), and watermelon crinkle leaf-associated virus 2 (WCLaV2). De novo assembly using SPAdes yielded complete genomes for both WSMoV isolates. For both isolates, the L segment was 8915 nt (GenBank accessions: FZNC, PQ310674; THBC, PQ310675) and the M segment was 4873 nt (GenBank accessions: FZNC, PQ310672; THBC, PQ310673). The S segment lengths were 3555 nt for FZNC (GenBank accessions: PQ310670) and 3558 nt for THBC (GenBank accessions: PQ310671).

### 3.4. Multiple-Sequence Alignment and Clustering of Complete Genome

Multiple-sequence alignments and clustering based on L-segment nucleotide sequences revealed that the ten WSMoV L segments shared 80.3–81.7% identity with the outgroup viruses peanut bud necrosis virus (PBNV) and watermelon bud necrosis virus (WBNV). The WSMoV L segments partitioned into two clades: Clade O (Old) comprised five isolates: three from Taiwan province, China (“Taiwan”, “TW”, and “PV0283”), one from Thailand (W2005) and one from Yunnan province, China (YNNP), with within-group identities of 97.2–99.7%. Clade N (New) comprised five mainland China isolates, including the two Jiangxi isolates (FZNC, THBC) and isolates from Guangxi (GL1), Guangdong (GZ), and Hebei (cucumber, SJZ), with within-group identities of 96.0–99.7%. The average sequence identity between the two groups was 85.4–86.0% (Figure 2A).

For the M segment, the ten WSMoV sequences exhibited 77.7–81.2% identity to the outgroups groundnut bud necrosis virus (GBNV) and WBNV. A similar two-group partition was observed: Clade N contained two mainland isolates (Guangdong GZ and Hebei SJZ) that were 98.5% identical to one another, whereas Clade O comprised the other eight isolates with internal identities of 95.8–99.7%. Inter-group identity for the M segment was 89.1–90.1% (Figure 2B).

Alignment of the S segment showed 79.1–79.8% identity to the outgroup GBNV. The S segment isolates again divided into Clades O and N. Clade O included 12 isolates, 9 from China (4 from Taiwan, 2 from Jiangxi, 2 from Yunnan, and 1 from Guangxi), 2 from Japan, and 1 from Thailand, with within-group identity of 93.4–99.5%. Clade N comprised five mainland isolates (two from Guangxi, one from Guangdong, one from Shandong, and one from Hebei) with internal identities of 96.6–99.7%. The between-group identity for the S segment was 88.1–88.8% (Figure 2C).

### 3.5. Phylogenetic Analysis

Phylogenetic trees were reconstructed from the full-length nucleotide sequences of the L (Figure 3A), M (Figure 3B), and S (Figure 3C) segments using the same isolate set employed for the multiple-sequence alignment. All three segment trees resolved two major clades (designated Clade O and Clade N), and the membership of isolates within each clade was concordant with the clustering results. In the M and S segment trees, Clade O contained a distinct subclade composed solely of the Yunnan peanut isolate YNNP. Notably, an incongruence in clade assignment was observed for three isolates: the two Jiangxi watermelon isolates (FZNC, THBC) and the Guangxi *Siraitia grosvenorii* isolate (GL1) grouped within Clade N in the L segment tree but clustered in Clade O in both the M and S segment trees. This topological discordance is indicative of segment reassortment among these isolates.

Phylogenetic reconstruction based on the nucleotide sequences of the *N* gene (39 isolates) produced a topology consistent with the complete genome analyses (Figure 4). Two clades (Clades O and N) were recovered: Clade N comprised 7 isolates (3 from Shandong, 1 from Hebei, 2 from Guangxi, and 1 from Guangdong), whereas Clade O comprised 32 isolates (13 from Thailand, 3 from Japan, 16 from China, 7 from Taiwan, 4 from Yunnan, 2 from Guangxi, 1 from Hainan, plus the 2 Jiangxi isolates described here). No clear association was observed between host species and phylogenetic grouping. The two Jiangxi watermelon isolates were most closely related to isolates from *Siraitia grosvenorii* (Guangxi) and melon (Hainan).

### 3.6. Recombination and Reassortment Analysis

Recombination screening was performed with RDP4 on the alignments of 10 L, 10 M, and 17 S segment genomes. No recombination signals were detected in the L or M segments. In contrast, a statistically supported recombination event (*p*-values from 10^−6^ to 10^−9^ detected by seven algorithms in Table 2) was identified in the S segment: the Banna2011 isolate (Yunnan) was inferred to be recombinant (Figure 5A), with breakpoints at nt 1173 (left breakpoint) and nt 1948 (right breakpoint). This event was supported by seven detection methods implemented in RDP4. The minor donor for the recombinant fragment (nt 1174–1947) was the Japanese watermelon isolates WSY and WSO, whereas the major donor, contributing nt 1–1173 and 1948–3554, was the isolates FZNC and THBC from Jiangxi and GL1 from Guangxi. Phylogenetic analyses were performed using two distinct genomic regions: one comprising nucleotides 1–1173 fused to 1948–3554 and the other spanning nt 1174–1947. When the phylogenetic tree was constructed using the former region, the recombinant isolate Banna2011 clustered together with Jiangxi isolates and one Guangxi isolate within a subclade (Appendix A). In contrast, phylogenetic analysis based on the latter region placed Banna2011 in a different subclade with two Japanese isolates (Appendix A). These discordant placements are consistent with a recombination event in the S segment identified by RDP.

To test for reassortment, L, M, and S segments from each isolate were concatenated into a pseudo-genome and re-analyzed in RDP4, treating inter-segment junctions as potential recombination breakpoints. This analysis revealed an exceptionally strong signal (*p*-value from 10^−14^ to 10^−151^ detected by seven algorithms in Table 2) located at approximately position 8900 of the pseudo-genome, which corresponds precisely to the L–M junction (Figure 5B). Three isolates, GL1 (from *Siraitia grosvenorii*, Guangxi) and the two Jiangxi watermelon isolates (FZNC and THBC), were implicated as reassortants. The inferred donors for this reassortment event were the Guangdong watermelon isolate GZ (source of the L segment; Clade N) and the Taiwanese tomato isolate PV0283 (source of the M and S segments; Clade O).

### 3.7. RT-PCR Validation of Reassortment

Segment-specific primers for the WSMoV L (WS-L4826F/WS-L5869R) and M (WS-M2347F/WS-M3278R) segments were used to amplify seven viral samples collected in 2023 and five collected in 2024 by RT-PCR. All amplicons were Sanger-sequenced; alignment to the corresponding regions of isolate FZNC1 yielded nucleotide identities >99.5% for the L segment and >99.4% for the M segment. Together with the previously reported >99.3% identity for the S segment (Section 3.2), these results confirm that all 12 Jiangxi isolates are highly similar and represent the reassortant strain described in this study.

## 4. Discussion

Since its first reports on watermelon in Japan in the 1980s [2] and on cucurbits in Taiwan in the 1990s, WSMoV occurrences in mainland China were historically confined to tropical regions such as Taiwan, Guangdong [5], Hainan [11], Guangxi [8,9], and Yunnan [7]. Recent records since 2022, however, document WSMoV presence in Zhejiang and Shandong [12], and a complete genome (isolate SJZ from Shijiazhuang City, Hebei Province) is now available in GenBank, indicating a trend toward spread into inland subtropical and temperate regions. Jiangxi Province, adjacent to Guangdong, lies in a subtropical zone and supports nearly 60,000 hectares of watermelon planting, making it a major production area in Eastern China. The detection of WSMoV and associated crop damage reported here therefore represents an expansion of the virus’s known distribution and highlights an increased risk of inland spread with a potential agronomic impact.

Reassortment is a common feature of segmented RNA viruses and occurs when a single host cell is co-infected by viruses of the same species but different genotypes, allowing for genomic segments from distinct parental strains to be mixed during replication and packaging [23]. A well-known example is influenza A virus, whose eight-segmented genome permits reassortment of haemagglutinin (HA) and neuraminidase (NA) segments and the rapid emergence of novel subtypes with altered host range and pandemic potential [24]. Evidence from *orthotospoviruses* likewise indicates that reassortment can shape population structure and adaptation: for instance, population-scale analyses of tomato spotted wilt virus (TSWV) identified reassortant genotypes in ~17% (38/224) of isolates, and reassortment has been implicated in TSWV’s rapid evolution, host adaptation, and the appearance of resistance-breaking variants [25]. Here, we report the first evidence for natural reassortment within the WSMoV population, involving exchange between distinct evolutionary clades. Experimental exploration of the biological impact of reassortment in WSMoV, such as its effects on host range, virulence, or vector compatibility, remains challenging owing to the scarcity of complete genomic sequences, the virus’s reliance on thrips for transmission (and lack of mechanical transmissibility), and the unavailability of infectious clones. Consequently, the evolutionary and biological significance of reassortment events remains unclear.

Previous studies [7,8,9,10,12,15] about phylogenetic analysis based on the nucleotide sequences of the *N* gene are broadly concordant with our results, which partition 39 WSMoV isolates into two major clades, designated N and O. Clade N is predominantly composed of isolates from China, whereas Clade O includes isolates from coastal and island regions (Thailand, Japan, and Taiwan) as well as a subset of mainland Chinese isolates from Yunnan, Guangxi, Hainan, and Jiangxi. The evolutionary scenario suggested by our phylogenetic results is that Clades N and O may have diverged from a common ancestor and subsequently evolved, at least in part, in distinct geographic/ecological contexts (Clade N in inland mainland China; Clade O in coastal/island and Southeast Asian regions). Mainland isolates that cluster within Clade O may represent more recent introductions, plausibly via trade or vector movement (thrips), into mainland China. The reassortant strains described here are consistent with co-infection by parental viruses related to a Taiwanese PV0283-like M/S donor (Clade O) and a Guangdong GZ-like L donor (Clade N), but this hypothesis remains tentative and will require testing with larger numbers of complete genome sequences and detailed epidemiological data. Although Japanese isolates form a distinct subclade within Clade O and appear to be relatively isolated, recombination analysis detected a fragment of the S segment in the Yunnan isolate Banna2011 that derives from the Japanese isolate WSY, indicating that limited genetic exchange between Japanese and Chinese lineages has occurred.

## 5. Conclusions

This study presents the first report of WSMoV in Jiangxi Province, expanding its known geographic range and highlighting the risk of further spread into China’s interior. We provide the first complete genome phylogenetic framework for WSMoV and uncover reassortment between major clades, offering new insights into the evolution and dissemination of WSMoV.

## Figures and Tables

**Figure 1 viruses-17-01448-f001:**
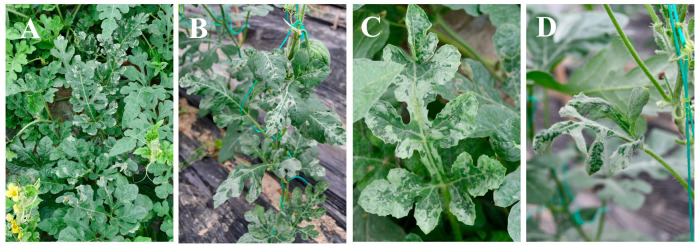
Symptoms of WSMoV infection in field-grown watermelon. (**A**,**C**) Samples collected from Nancheng County, Jiangxi Province; (**B**,**D**) samples collected from Taihe County, Jiangxi Province.

**Figure 2 viruses-17-01448-f002:**
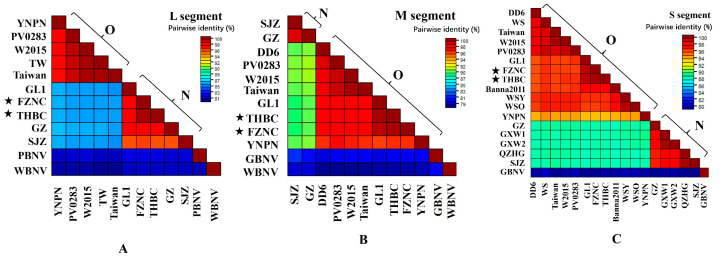
Pairwise precent-identity matrix for L, M, and S segments. (**A**) L segment, (**B**) M segment, (**C**) S segment. The isolates marked with a star (★) indicate the Jiangxi isolates in this study.

**Figure 3 viruses-17-01448-f003:**
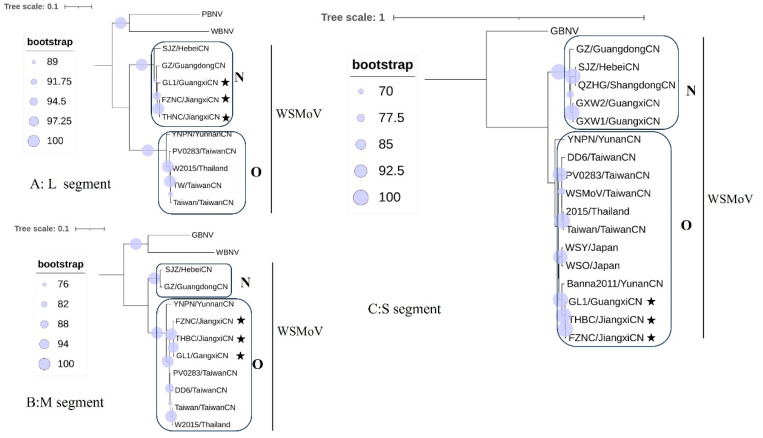
Phylogenetic trees of the nucleotide sequence of L, M, and S segments. (**A**) L segment, (**B**) M segment, (**C**) S segment. The isolates marked with a star (★) indicate those involved in a potential reassortment event; nodes marked with blue circles represent the clade supported by a bootstrap value more than 70%. Peanut bud necrosis virus (PBNV), watermelon bud necrosis virus (WBNV), and groundnut bud necrosis virus (GBNV) were used as outgroups for L, M, and S trees, respectively.

**Figure 4 viruses-17-01448-f004:**
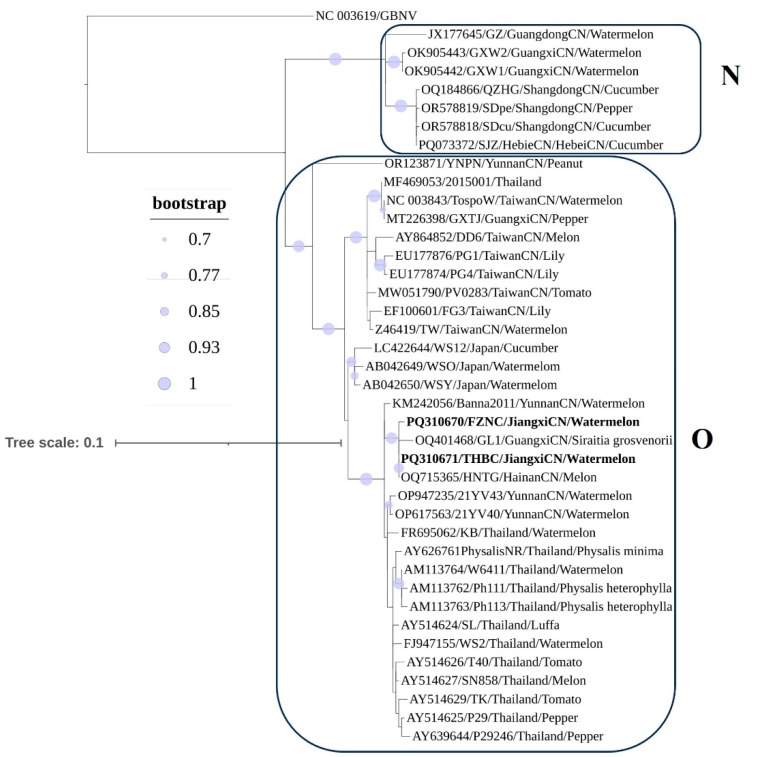
Phylogenetic tree based on nucleotide sequence of *N* gene. Isolates marked from Jiangxi Province in this study are highlighted in **bold**. Nodes marked with blue circles indicate clades supported by bootstrap >70%. Accession number, isolate name, collected location, and host are shown at each node. Groundnut bud necrosis virus (GBNV) was used as the outgroup.

**Figure 5 viruses-17-01448-f005:**
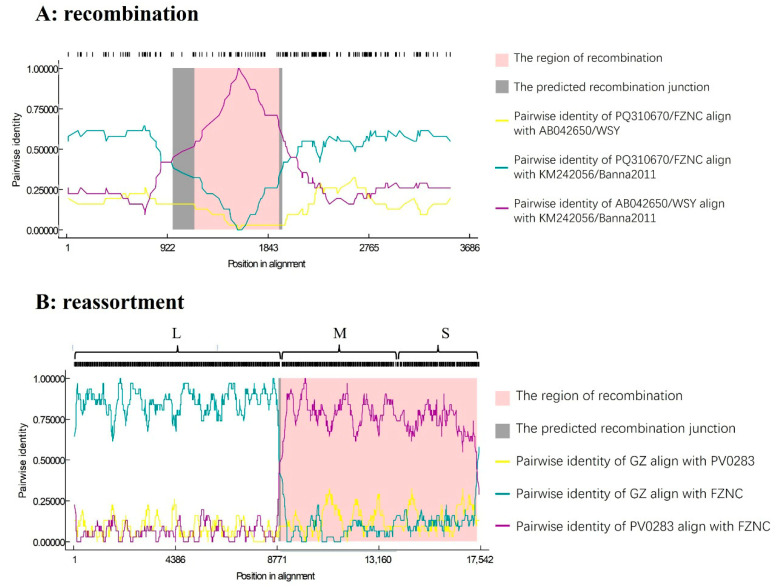
Recombination and reassortment analysis of the WSMoV genomes using RDP. (**A**) Nucleotide sequence identity along the genomes of FZNC, banna2011, and WSY. (**B**) Nucleotide sequence identity along the genomes of GZ, FZNC, and PV0283.

**Table 1 viruses-17-01448-t001:** Details of the WSMoV isolates used in this study.

Isolate	Location	Year	Host	GenBank Accessions
FZNC	Jiangxi, China	2023	WM	L: PQ310674, M: PQ310672, S: PQ310670
THBC	Jiangxi, China	2023	WM	L: PQ310675, M: PQ310673, S: PQ310671
GZ	Guangdong, China	2011	WM	L: JX177647, M: JX177646, S: JX177645
SJZ	Hebei, China	2023	Cu	L: JX177647, M: PQ073373, S: PQ073372
YNPN	Yunnan, China	2022	Pe	L: OR123870, M: OR123869, S: OR123871
PV0283	Taiwan, China	2020	To	L: MW051788, M: MW051789, S: MW051790
W2015	Thailand	2015	NC	L: MF469051, M: MF469052, S: MF469053
Taiwan	Taiwan, China	1988	WM	L: NC_003832, M: NC_003841, S: NC_003843
GL1	Guangxi, China	NC	SG	L: OQ401466, M: OQ401467, S: OQ401468
TW	Taiwan, China	1992	M	L: AY863200,
QZHG	Shandong, China	2022	Cu	S: OQ184866
GXW1	Guangxi, China	2016	WM	S: OK905442
GXW2	Guangxi, China	2016	WM	S: OK905443
DD6	Taiwan, China	NC	M	M: AY864852, S: DQ157768
WS	NC	NC	NC	S: Z46419
WSO	Japan	1982	WM	S: AB042649
WSY	Japan	1982	WM	S: AB042650
Banna2011	Yunnan, China	2011	WM	S: KM242056

WM represents *Citrullus lanatus*, Cu represents *Cucumis sativus*, Pe represents *Arachis hypogaea*, To represents *Solanum lycopersicum*, SG represents *Siraitia grosvenorii*, and M represents *Cucumis melo*. NC indicates that host information is not recorded in GenBank.

**Table 2 viruses-17-01448-t002:** Recombination and reassortment events detected in the WSMoV genome using the seven methods in the RDP4 program.

Methods of Detection	*p*-Value	
	Recombination	Reassortment
RDP	4.93 × 10^−6^	1.35 × 10^−151^
GENECONV	4.14 × 10^−9^	6.09 × 10^−136^
BootScan	4.18 × 10^−6^	1.96 × 10^−134^
MaxChi	1.07 × 10^−7^	5.18 × 10^−62^
Chimaera	2.04 × 10^−6^	2.74 × 10^−64^
SiScan	5.82 × 10^−10^	1.37 × 10^−68^
3Seq	7.14 × 10^−6^	3.73 × 10^−14^

## Data Availability

The original contributions presented in the study are included in the article. Further inquiries can be directed to the author.

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
