# Peer review of "Identification of Reassortment of Orthotospovirus citrullomaculosi in Jiangxi Province, China"

_viruses, 2025, doi:10.3390/v17111448_

Round 1
Reviewer 1 Report
Comments and Suggestions for Authors
Peng et al. report their work on the identification and genomic analysis of an important orthotospovirus on watermelon. They surveyed the disease incidence and identified the virus by RT-PCR and high-throughput sequencing. Two full genome sequences were obtained from two isolates of Jiangxi Province, China. Genomic and phylogenetic analysis was perforemd and they found reassortment between isolates of China. This work is very important for knowing the epidermic of this important tospovirus on curcubits. Their data provide essential information for occurence and evolution of this virus.
Minor corrections list below.
L43 Tospoviridae should be italic.
L48 physalis. Capital the first letter.
L60-62 virus is not diease. Reword the sentence, describe the symptoms in concise observed in fields.
L87-93 Were the primers designed by the authors? If yes, give the information on their locations on genome and product size. If not, give the references.
L97 Capital the first letter of complete
L119 move the reference number before dot.
L144-150 Move the first sentence after the symptoms. Move the last sentence to Discussion section.
Author Response
Peng et al. report their work on the identification and genomic analysis of an important orthotospovirus on watermelon. They surveyed the disease incidence and identified the virus by RT-PCR and high-throughput sequencing. Two full genome sequences were obtained from two isolates of Jiangxi Province, China. Genomic and phylogenetic analysis was perforemd and they found reassortment between isolates of China. This work is very important for knowing the epidermic of this important tospovirus on curcubits. Their data provide essential information for occurence and evolution of this virus.
Dear reviewer:
Thank you for your time and effort in handling our manuscript titled “Identification of Reassortment of Orthotospovirus citrullomaculosi in Jiangxi Province, China” (viruses-3903258). We have revised our manuscript in light of the detailed and helpful comments of the reviewers. We highlighted major changes in the revised manuscript with red text.
Attached please find our detailed responses to the specific comments of the reviewers. We hope that we have addressed the reviewers’ concerns and this revised manuscript is suitable for publication in Viruses.
Minor corrections list below.
The whole manuscript has been refined by native English speaker.
L43 Tospoviridae should be italic.
Response: Corrected.
L48 physalis. Capital the first letter.
Response: Corrected
L60-62 virus is not diease. Reword the sentence, describe the symptoms in concise observed in fields.
Response: Thanks for the suggestion. The sentence has now been reworded (line 78-80 in revised manuscript).
.
L87-93 Were the primers designed by the authors? If yes, give the information on their locations on genome and product size. If not, give the references.
Response: All the primers used in this study were designed by the authors based on the reference genome. According to the reviewers' suggestions, in the supplementary materials, Table S1 provides all the primer information, including the reference genome, location, sequences and product size.
L97 Capital the first letter of complete
Response: Corrected
L119 move the reference number before dot.
Response: Corrected
L144-150 Move the first sentence after the symptoms. Move the last sentence to Discussion section.
Response: Thanks for the suggestion. The first sentence has been moved after the symptoms. the last sentence has been deleted.
Reviewer 2 Report
Comments and Suggestions for Authors
In this study, the authors discovered two new isolates of WSMoV infecting watermelon in Jiangxi province, and found that there is recombination and reassortment in these two isolates with other WSMoV isolates. Overall, the study is well designed and well-written. But still, I have some questions as below.
- Line 50-51: Missed a reference, WSMoV was also found in cucumber, squash, and whitegourd in Guangdong, cite this reference: Li, Z. G., Nong, Y., Farooq, T., Tang, Y. F., She, X. M., Yu, L., ... & He, Z. F. (2022). Small RNA deep sequencing reveals the presence of multiple viral infections in cucurbit crops in Guangdong, China. Integr. Agric., 21(5), 1389-1400.
- Figure 2, why the two Jiangxi isolates were classified in different clade by L, M, and S, can authors explain the reason? Is it common in other viruses?
- In the recombination analysis, the authors found that “The minor donor for the recombinant fragment was the Japanese watermelon isolate WSY, whereas the major donor, contributing nt 1–971 and 1284–3554, was the isolate FZNC, THBC from Jiangxi and GL1 from Guangxi”. It is better to show the similarity analysis like the authors did in the reassortment analysis.
- Line 267, what is the identity to the homologous region of the GZ isolate?
Author Response
In this study, the authors discovered two new isolates of WSMoV infecting watermelon in Jiangxi province, and found that there is recombination and reassortment in these two isolates with other WSMoV isolates. Overall, the study is well designed and well-written. But still, I have some questions as below.
Dear reviewer:
Thank you for your time and effort in handling our manuscript titled “Identification of Reassortment of Orthotospovirus citrullomaculosi in Jiangxi Province, China” (viruses-3903258). We have revised our manuscript in light of the detailed and helpful comments of the reviewers. We highlighted major changes in the revised manuscript with red text.
Attached please find our detailed responses to the specific comments of the reviewers. We hope that we have addressed the reviewers’ concerns and this revised manuscript is suitable for publication in Viruses.
Line 50-51: Missed a reference, WSMoV was also found in cucumber, squash, and whitegourd in Guangdong, cite this reference : Li, Z. G., Nong, Y., Farooq, T., Tang, Y. F., She, X. M., Yu, L., ... & He, Z. F. (2022). Small RNA deep sequencing reveals the presence of multiple viral infections in cucurbit crops in Guangdong, China. Integr. Agric., 21(5), 1389-1400.
Response: We are sorry for the miss of reference. The reference has been added to reference [10] (Line 58-58 in revised manuscript).
Figure 2, why the two Jiangxi isolates were classified in different clade by L, M, and S, can authors explain the reason? Is it common in other viruses?
Response: Figure 2 presents the results of the multiple sequence alignment and clustering of the complete genomes. It indicates that for the two Jiangxi isolates, the L segment clusters differently from the M and S segments in the nucleotide sequence identity analysis. The reason for this discrepancy is a key finding of our study: the two Jiangxi isolates have undergone genomic reassortment. Reassortment is a common phenomenon in multi-segmented RNA viruses, and we have discussed examples of this from other viral systems in the Discussion section (Lines 350-355 in revised manuscript).
In the recombination analysis, the authors found that “The minor donor for the recombinant fragment was the Japanese watermelon isolate WSY, whereas the major donor, contributing nt 1–971 and 1284–3554, was the isolate FZNC, THBC from Jiangxi and GL1 from Guangxi”. It is better to show the similarity analysis like the authors did in the reassortment analysis.
Response: We thank the reviewer for this comment. To further validate the recombination event, we conducted separate phylogenetic analyses on two distinct genomic regions: one comprising nucleotides 1–1173 fused to 1949–3554, and the other spanning nt 1174–1949. The results from these analyses, which confirm the recombination, are presented in Lines 289-296 (in revised manuscript) and Supplementary Figure S2. We have also corrected errors related to the initial recombination analysis in the revised manuscript (Lines 283-286 in revised manuscript), and we kindly ask the reviewers to draw their attention to this update.
Line 267, what is the identity to the homologous region of the GZ isolate?
Response: "The homologous region of Taiwan reference L segment" refers to the nucleotide sequence of the homologous region in the genome of the Taiwan isolates, as determined by the amplification product of primers WS-L4826F/WS-L5869R. We have rewritten the sentence I rephrased this sentence to make the meaning clearer (Line 314-2316in revised manuscript).
Reviewer 3 Report
Comments and Suggestions for Authors
The present article by Peng et al., has been written well. The abstract section needs to be more concise; introduction can be elaborative and include economic importance of this disease. Total number of symptomatic/asymptomatic sample collection and area infested need to be explained. Data set used in this study is very small. Although this article is interesting, it needs to be revised carefully. In my opinion, this paper might get accepted after incorporating these changes.
Major revision;
Line no. 15; Do not abbreviate in abstract section.
Line no. 17-18; Use proper terminology if using isolate or stain! Just make sure it is correct.
Line no. 16-21; This sentence is confusing, reframe it.
Line no. 32; There are different species of Thrips palmi which one you are referring to?
Line no. 46-59; What is the biological significance of this virus?
Line no. 63-64; molecular biological?
Line no. 69; full-length genome….
Line no. 75; The area observed for disease assessment is very small for field study. Why have authors selected only 0.67 hectares?
Line no. 75-76; Only one year of data is not significant!
Line no. 78-80; Total how many plants were assessed (including both the season)?
Line no. 88-89; Primers were targeting which region?
Line no. 107-109; What was the full-length size of L, S, and M segment?
Line no. 117; How many N-gene sequences were amplified in this study? Why do the authors choose only N-gene?
Line no. 144; Disease spread was in more than two counties then why authors have chosen to collect data from Nancheng and Taihe county? Are these places major watermelon growing counties? However, disease incidence is not even 50%!
Line no. 216; M and Segment of GL1/THBC/ FZNC isolates showing more relatedness with O-clade of WSMoV whereas, L-segment are in N-clade. However, number of sequences selected here is too less!
Line no. 271; Table 1 what is ‘NC’?
Author Response
The present article by Peng et al., has been written well. The abstract section needs to be more concise; introduction can be elaborative and include economic importance of this disease. Total number of symptomatic/asymptomatic sample collection and area infested need to be explained. Data set used in this study is very small. Although this article is interesting, it needs to be revised carefully. In my opinion, this paper might get accepted after incorporating these changes.
Dear reviewer:
Thank you for your time and effort in handling our manuscript titled “Identification of Reassortment of Orthotospovirus citrullomaculosi in Jiangxi Province, China” (viruses-3903258). We have revised our manuscript in light of the detailed and helpful comments of the reviewers. We highlighted major changes in the revised manuscript with red text.
Attached please find our detailed responses to the specific comments of the reviewers. We hope that we have addressed the reviewers’ concerns and this revised manuscript is suitable for publication in Viruses.
Major revision;
Line no. 15; Do not abbreviate in abstract section.
Response:Corrected
Line no. 17-18; Use proper terminology if using isolate or stain! Just make sure it is correct.
Response: The difference between an isolate and a strain, as I understand it, is as follows: An Isolate is an operational and descriptive term that addresses the question of "where does it come from?" A Strain, on the other hand, is a taxonomic and characteristic term that answers the question of "what is it?" Accordingly, in the manuscript, "isolate" is employed for geographical context, and "strain" in discussions of virological properties like recombination or reassortment.
Line no. 16-21; This sentence is confusing, reframe it.
Response: Thanks for the suggestion. The sentence has now been reworded (Line 14-27 in revised manuscript).
Line no. 32; There are different species of Thrips palmi which one you are referring to?
Response: To our knowledge, Thrips palmi (commonly known as the melon thrips) is a species belonging to the genus Thrips (order: Thysanoptera).
Line no. 46-59; What is the biological significance of this virus?
Response: We thank the reviewer for this comment. In the second paragraph of the Introduction, the original manuscript described the biological significance of WSMoV and the symptoms it causes in different hosts. In the revised manuscript, we have expanded this section to address the virus’s potential economic impact, its recent geographic range expansion, and emerging prevalence trends. (Line 66-76 in revised manuscript)
Line no. 63-64; molecular biological?
Response: We sorry for the unclear wording. To improve clarity, we have replaced “molecular biological” with “RT-PCR.” (Line84 in revised manuscript)
Line no. 69; full-length genome….
Response: Thanks for the suggestion. The sentence has now been reworded.(Line 90-91 in revised manuscript)
Line no. 75; The area observed for disease assessment is very small for field study. Why have authors selected only 0.67 hectares?
Response: In Jiangxi Province, where the terrain is predominantly hilly and mountainous, flat and contiguous land is generally reserved for cereal crops such as rice. In contrast, commercial crop production is primarily undertaken by smallholder families. These farms are typically scattered and fragmented, and large-scale monocultures of cash crops like watermelon are uncommon. Most operations cover less than one hectare. The two cultivation plots examined in this study exemplify this pattern. Furthermore, surveys conducted at another farm approximately 10 km away in Nancheng County detected no WSMoV infection.
Line no. 75-76; Only one year of data is not significant!
Response:We thank the reviewer for this important point and agree that a single growing season of survey data limits inferences about disease epidemiology. Nonetheless, the primary finding of our study,the identification of the Jiangxi WSMoV isolates as reassortant strains,remains robust and is not undermined by the survey’s temporal or sample-size limitations. In fact, we have conducted additional field investigations. The Nancheng-site could not be resampled for watermelon in the second growing season because the farmer planted Chinese cabbage that season. In Taihe County, the second growing season occurred in spring 2024; detect of watermelon plants from that farm were negative for WSMoV. Importantly, in 2024 we collected five WSMoV-positive watermelon samples from three other Jiangxi counties (Guangchang, Yudu, and Huichang); RT-PCR validation showed that all five of these isolates are also reassortant strains. We have incorporated these supplementary results into the final paragraph (Line 289-296) of the Results section in the revised manuscript.
Line no. 78-80; Total how many plants were assessed (including both the season)?
Response: Disease assessment, carried out over one growing season, comprised approximately 50 plants per plot.
Line no. 88-89; Primers were targeting which region?
Response: All the primers used in this study were designed by the authors based on the reference genome. According to the reviewers' suggestions, in the supplementary materials, table S1 provides all the primer information, including the reference genome, location, sequences and product size.
Line no. 107-109; What was the full-length size of L, S, and M segment?
Response: the full-length size of L, S, and M segment shown in “Results” section 3.3 (Line 210-213 in revised manuscript).
Line no. 117; How many N-gene sequences were amplified in this study? Why do the authors choose only N-gene?
Response: Twelve N-gene sequences were amplified in this study. The information was mentioned at section 3.2 (Line 190-203) in revised manuscript. As noted in Line 46-47 in revised manuscript, the N-gene serves as the criterion for species demarcation and represents the most frequently amplified and characterized region in previous studies. Our decision to focus on this genomic region facilitates comparison with previous studies.
Line no. 144; Disease spread was in more than two counties then why authors have chosen to collect data from Nancheng and Taihe county? Are these places major watermelon growing counties? However, disease incidence is not even 50%!
Response:The occurrence of the disease was not limited to these two counties. Field surveys in 2023 and 2024 detected sporadic cases (<5%) with mild symptoms in watermelon plots in other counties, which did not cause noticeable yield loss or reductions in marketability. The watermelon planting area in Nancheng and Taihe counties is roughly 500 hectares, so they are not major watermelon-producing counties in Jiangxi Province. In this study the disease incidences in the two plots were 46% and 24%, respectively. Although the incidence did not exceed 50%, the infection may have occurred at an early growth stage and already substantially affected yield and marketability, which drew growers’ attention and led them to request our investigation and pathogen identification. The fact that observed incidence did not exceed 50% may be explained by atypical or latent symptoms; the true infection rate may be higher than the recorded disease incidence. The disease survey conducted here was a simple assessment rather than a formal epidemiological investigation.
Line no. 216; M and S segment of GL1/THBC/ FZNC isolates showing more relatedness with O-clade of WSMoV whereas, L-segment are in N-clade. However, number of sequences selected here is too less!
Response: In addition to the two Jiangxi isolates analyzed in this study, only seven WSMoV isolates in GenBank possess complete genome sequences encompassing the L, M, and S segments. However, S-segment sequences are available for 15 isolates. The relevant information is summarized in Table 1.
Line no. 271; Table 1 what is ‘NC’?
Response: We apologize for not explaining “NC” in Table 1. “NC” denotes that host information for the isolate is not recorded in GenBank. This definition has been added to Table 1 of the revised manuscript.
Round 2
Reviewer 3 Report
Comments and Suggestions for Authors
After revision, MS has been significantly improved. I endorsed it for publication. Thank You.